# ABACO: A New Model of Microalgae-Bacteria Consortia for Biological Treatment of Wastewaters

Ana Sánchez-Zurano [1,*], Enrique Rodríguez-Miranda [2], José Luis Guzmán [3], Francisco Gabriel Acién-Fernández [1], José M. Fernández-Sevilla [1] and Emilio Molina Grima [1]

[1] Department of Chemical Engineering, University of Almería, Ctra. Sacramento s/n, ceiA3, CIESOL, 04120 Almería, Spain; facien@ual.es (F.G.A.-F.); jfernand@ual.es (J.M.F.-S.); emolina@ual.es (E.M.G.)

[2] Department of Mechanical and Industrial Engineering, University of Brescia, Via Branze 38, 25123 Brescia, Italy; e.rodriguezmiran@unibs.it

[3] Department of Informatics, University of Almería, Ctra. Sacramento s/n, ceiA3, CIESOL, 04120 Almería, Spain; joseluis.guzman@ual.es

[*] Correspondence: asz563@ual.es

**Abstract:** Microalgae-bacteria consortia have been proposed as alternatives to conventional biological processes to treat different types of wastewaters, including animal slurry. In this work, a microalgae-bacteria consortia (ABACO) model for wastewater treatment is proposed, it being calibrated and validated using pig slurry. The model includes the most relevant features of microalgae, such as light dependence, endogenous respiration, and growth and nutrient consumption as a function of nutrient availability (especially inorganic carbon), in addition to the already reported features of heterotrophic and nitrifying bacteria. The interrelation between the different populations is also included in the model, in addition to the simultaneous release and consumption of the most relevant compounds, such as oxygen and carbon dioxide. The implementation of the model has been performed in MATLAB software; the calibration of model parameters was carried out using genetic algorithms. The ABACO model allows one to simulate the dynamics of different components in the system, and the relative proportions of microalgae, heterotrophic bacteria, and nitrifying bacteria. The percentage of each microbial population obtained with the model was confirmed by respirometric techniques. The proposed model is a powerful tool for the development of microalgae-related wastewater treatment processes, both to maximize the production of microalgal biomass and to optimize the wastewater treatment capacity.

**Keywords:** microalgae; bacteria; modelling; wastewater treatment; nutrients; photobioreactor

## 1. Introduction

One of the most critical environmental challenges of the 21st century envisaged by humanity is the expansion of the population, which will result in increased urban wastewater production [1] and large amounts of animal slurry caused by the rise in meat production [2,3]. The world's growing population, along with (i) a rapid industrialization, (ii) intensive agriculture, (iii) the effluent discharged below an environmentally safe level, and (iv) the lack of technologies to reclaim used water could lead to a scarcity of clean water in many countries [4]. The current conventional wastewater treatment methods have become quickly outdated because they need a lot of land, intensive energy input, and a lot of money [5]. As an alternative strategy to beat these disadvantages, microalgae-based wastewater treatment is gaining an increased importance in the context of European bioeconomy, because of its potential to treat wastewater, recover nutrients of wastewater, and produce a large variety of valuable compounds with applications in agriculture, aquaculture, and food production, among others [6–8]. The use of microalgae for wastewater treatment involves the emergence of complex microalgae–bacteria consortia which vary as functions of environmental and operational conditions [9].

Microalgae are photosynthetic microorganisms that grow using inorganic carbon ($CO_2$) as a carbon source, and light as an energy source. During this growth, microalgae release oxygen which can use by heterotrophic bacteria to oxidate the organic matter present in influent wastewater. At the same time, heterotrophic bacteria supply $CO_2$ for photosynthetic activity, completing the cycle. Besides, the oxygen produced by microalgae can be used by nitrifying bacteria to oxidize the ammonium to nitrate (nitrification process), consuming $CO_2$ as a carbon source too [10–12]. Since microalgae–bacteria consortia in wastewater treatment was described in 1953 by [13], multiple microalgae–bacteria models have been described and validated [14–17]. These mathematical models offer great appeal to studying microalgae–bacteria interactions because they can provide useful tools for design and control purposes, in addition to model simulators, which can all lead to an increase the process efficiency [18].

In most of the proposed mathematical models, the part related to the activity of the bacteria is widely obtained and validated through the Activated sludge models (ASM) [19]. However, information on microalgae parameters in wastewater treatment systems is scarce. Therefore, in this work, a new microalgae–bacteria mathematical model named ABACO is proposed; the characteristic parameters of microalgae in it were obtained experimentally in previous works [20,21]. Thus, the main purpose of this study was to develop, calibrate, and validate the whole microalgae and bacteria model with experimental data from duplicate laboratory-scale photobioreactors using pig slurry as a nutrient source. The implementation of the microalgae–bacteria model has been performed in MATLAB software, and it allows one to simulate the dynamics of different components in the system and the relative proportions of microalgae and bacteria. Moreover, the model has a series of parameters whose exact values are unknown, being within a range. The calibration of these parameters has been carried out using genetic algorithms, which allow determining their values from minimizations of given cost functions. This calibration procedure provides a simple and fast adjustment method for the characterization of the model parameters, even allowing recalibration with different scenarios in a very easy way, such as for different strains and culture mediums. Moreover, notice that thanks to the proposed calibration process, it is possible to estimate the percentage of each species in the reactor, which is also a relevant contribution of the methodology proposed in this work.

## 2. Materials and Methods

### 2.1. Microorganisms and Culture Conditions

The microalgal strain used to inoculate the photobioreactors was *Scenedesmus almeriensis*. The stock culture of *Scenedesmus almeriensis* was maintained photo-autotrophically in spherical flasks (1 L capacity) using the Arnon medium [22]. The microalgal culture was continuously bubbled with $CO_2$-enriched air (1%), which allowed us to control the pH at 8.0. The air temperature in the chamber was controlled in order to obtain a desire temperature (22 °C). The culture temperature was set at 25 °C, controlled by regulating the air temperature in the chamber. The culture was artificially illuminated in a 12:12 h L/D cycle using four Philips PL-32W/840/4p white-light lamps, providing an irradiance of 750 µE/m$^2$ s on the spherical 1.0 L flask surface. Two laboratory-scale photobioreactors were inoculated using the culture stock. The average composition of the Arnon medium used is reported in Table 1.

### 2.2. Laboratory Photobioreactors

Two hand-made photobioreactors made with polymethylmethacrylate (0.08 m in diameter, 0.2 m in height and with a 1 L capacity) were used to perform the experiments (Figure 1). The reactors were inoculated with 20% of *Scenedesmus almeriensis* and diluted pig slurry (20%). The photobioreactors were operated in the laboratory but simulating outdoor conditions prevailing in outdoor raceway reactors. Firstly, the photobioreactors were operated in batch mode for 5 days to obtain a high biomass concentration. Afterwards, they were operated in continuous mode by removing 20% of the culture every day and replacing

it with fresh piggery wastewater. The dissolved oxygen in the culture was controlled below 200 % Sat to avoid negative effects because of excessive dissolved oxygen accumulation. For that, air was supplied on demand. Additionally, the pH was controlled at 8.0 using $CO_2$ injections. To simulate the outdoor solar cycle, the reactors were artificially illuminated using eight 28 W fluorescent tubes (Philips Daylight T5). The maximum irradiance (PAR) inside the reactors without cells was 1000 $\mu Em^{-2} s^{-1}$, measured using an SQS-100 spherical quantum sensor (Walz GmbH, Effeltrich, Germany). The culture temperature was kept at 25 °C by controlling the temperature of the culture chamber in which the photobioreactors were located. The average composition of the piggery wastewater used is reported in Table 1.

**Table 1.** Average compositions of the culture medium and piggery wastewater used as the influent in the bioreactors. Concentrations expressed as mg $\times L^{-1}$.

| Parameters | Piggery Wastewater | Arnon Medium |
|---|---|---|
| pH | $8.1 \pm 0.3$ | $7.5 \pm 0.2$ |
| COD | $2181.7 \pm 100.9$ | $16.0 \pm 1.2$ |
| Nitrogen-Nitrate | $56.5 \pm 2.7$ | $140.0 \pm 4.5$ |
| Chloride | $2060.2 \pm 23.5$ | $78.9 \pm 2.1$ |
| Potassium | $1800 \pm 1.6$ | $325.1 \pm 6.3$ |
| Calcium | $350.1 \pm 0.2$ | $364.9 \pm 5.5$ |
| Magnesium | $108.2 \pm 14.1$ | $12.2 \pm 0.6$ |
| Phosphorus-Phosphate | $119.2 \pm 5.1$ | $39.3 \pm 3.1$ |
| Nitrogen-Ammonium | $1495.6 \pm 17.7$ | $0.0 \pm 0.1$ |
| Iron | $4.8 \pm 0.01$ | $5.0 \pm 0.3$ |
| Copper | $1.1 \pm 0.1$ | $0.02 \pm 0.00$ |
| Manganese | $2.6 \pm 0.0$ | $0.5 \pm 0.02$ |
| Zinc | $20.1 \pm 0.2$ | $0.06 \pm 0.01$ |
| Boron | $5.3 \pm 0.1$ | $0.4 \pm 0.0$ |

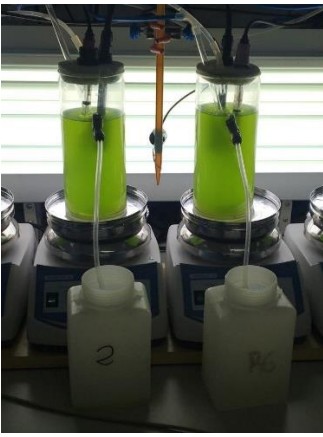

**Figure 1.** Laboratory photobioreactors used for performing the experiments.

*2.3. Biomass Concentration and Analytical Methods*

The biomass concentration (Cb) was measured by dry weight. For that, aliquots (100 mL) of each photobioreactors the culture were filtered through the Macherey–Nagel MN 85/90 glass fiber filters. Then, the filters were dried in an oven at 80 °C for 24 h. Standard official methods were used to analyze the composition of the piggery wastewater and the supernatants from microalgae–bacteria cultures. The phosphate was measured by visible spectrophotometry through the phospho-vanado-molybdate complex (phosphate standard for IC: 38364). The nitrate was quantified by measuring optical density at 220 nm and 275 nm (nitrate Standard for IC: 74246). The ammonium was measured according to the Nessler method (ammonium standard for IC: 59755). The chemical oxygen demand (COD) was determined by spectrophotometric measurement using Hach–Lange kits (LCl-400).



### 2.4. Model Calibration and Validation

MATLAB Software was used to carry out the model calibration process using genetic algorithms through the Genetic Algorithm Optimization Toolbox (GAOT), based on [23]. Additionally, the model validation with experimental data was performed using MATLAB Software.

### 2.5. Respirometry: Measurements of the Photosynthesis and Respiration Rates

In order to validate experimentally the percentage of each microbial population proposed in the biological model, respirometric measurements were performed when at steady state. The percentages of microalgae and bacteria in the culture were estimated as functions of the microalgae net photosynthesis rate, the heterotrophic respiration rate, and the nitrifying respiration rate, respectively. These measurements were performed with handmade photo-respirometer equipment. This equipment is described in detail in [11]. The method allows one to determine the photosynthesis and respiration rates through the variations in dissolved oxygen concentrations in microalgae–bacteria cultures, as described in detail in [11].

For evaluating the microalgae net photosynthesis rate of each microalgae–bacteria culture, a sample of the culture was exposed to four light–dark cycles of four minutes each to measure and register the variation in dissolved oxygen. During the light phases, the photosynthetic microalgae generated dissolved oxygen, and this dissolved oxygen was consumed by the endogenous respiration during the dark periods. Thus, the microalgae net photosynthesis rate was calculated as the difference between the slope of the oxygen production during the light period and the slope of the oxygen consumption during the dark period. Subsequently, another sample of the culture was used to determine the heterotrophic respiration rate. For this purpose, 0.8 mL of sodium acetate (30 g/L) was added to the sample and it was exposed to four light–dark cycles of 4 min each. The respiration rate of the heterotrophic bacteria was calculated as the slope of the oxygen consumption with sodium acetate minus the slope of the oxygen consumption during the dark period in the endogenous culture. By following the same method, another sample was used to measure the nitrifying respiration rate of the culture. However, the nitrifying activity was determined using 0.8 mL of ammonium chloride (3 g/L) instead of sodium acetate. The respiration rate of the nitrifying bacteria was calculated as the slope of the oxygen consumption with ammonium chloride minus the slope of the oxygen consumption during the dark period in the endogenous culture [11].

Finally, in order to correct the influence of oxygen desorption on the photo-respirometric measurements, the oxygen mass transfer coefficient ($K_La$) was calculated. This coefficient was measured in the system according to Equation (1).

$$\frac{dX_{O_2}}{dt} = K_La \left( X_{O_2}^* - X_{O_2} \right) \tag{1}$$

where $\frac{dX_{O_2}}{dt}$ is the oxygen accumulation expressed as the derivate of $X_{O_2}$ (mg/L) concentration over time, $K_La$ is the global oxygen mass transfer coefficient ($h^{-1}$), and $X_{O_2}^*$ is the oxygen saturation concentration in the liquid. Further detailed descriptions of the equipment, the standard protocol, and the metabolic rate calculations are in [11].

## 3. Results

This section, divided into four parts, presents the results obtained for the joint model of microalgae biomass production combined with pig slurry treatment. The first part provides a description of the mass balances of the model related to the process. The second part shows the mathematical background relative to the growth rate of the species involved. The third part shows the calibration process and the results. Finally, in the fourth part, the validation results obtained for the model are presented.

### 3.1. Model Concept

In microalgae-based wastewater treatment, different types of microbial consortia appear as a function of environmental and operational conditions. Figure 2 shows the biological process taking place in the reactor when using wastewater (i.e., diluted pig slurry) as the culture medium. Under illumination, microalgae ($X_{ALG}$) fix carbon dioxide ($CO_2$) and release oxygen ($O_2$) while assimilating nutrients, such as ammonium ($NH_4$), nitrate ($NO_3$), and phosphate ($PO_4$). The $O_2$ produced by the photosynthesis is essential for the degradation of the biodegradable soluble organic matter (BSOM) by heterotrophic bacteria ($X_{HET}$), BSOM being a fraction of total organic matter (COD) contained in wastewater. In turn, during bacterial oxidation of soluble organic matter, $CO_2$ is produced, it being available for photosynthesis and the nitrification process. During nitrification, nitrifying bacteria ($X_{NIT}$) transform $NH_4$ already contained at the inlet culture medium into $NO_3$, while also consuming $O_2$ produced through photosynthesis.

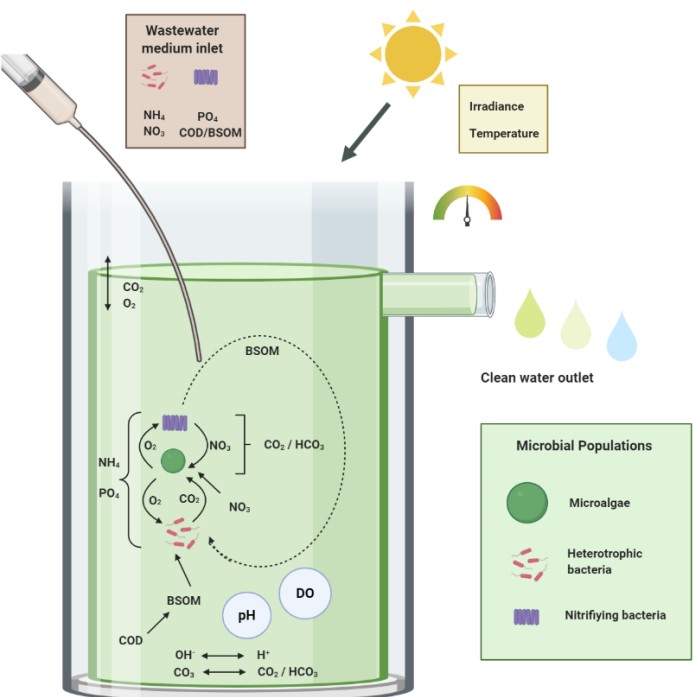

**Figure 2.** Biological process for microalgae biomass production coupled with wastewater treatment.

The developed model includes the mass balances of major compounds involved into the biological process, in addition to the growth rate of the different species involved (microalgae and bacteria) as a function of culture conditions and nutrients availability. Starting from known initial conditions and variables already measured in the reactor it is possible to simulate the evolution of the system over time, thus the variation of both compounds and microorganisms. Figure 3 shows the most relevant inputs and outputs of the model, and the initial conditions required. The inputs for the model are the variables commonly measured in photobioreactors such as irradiance, dissolved oxygen, pH and temperature. The model outputs are the concentrations of major microorganisms already considered such as microalgae, heterotrophic bacteria and nitrifying bacteria; in addition to the concentration of major components and nutrients involved into the biological process such as oxygen, carbon dioxide, total inorganic carbon, ammonium, nitrate, phosphate, and BSOM. For the right estimation of the evolution of the system it is necessary to establish values for the initial conditions, which correspond to the initial concentrations of the nutrients and the total biomass, the initial percentages of species in the photobioreactor and the calibration parameters.

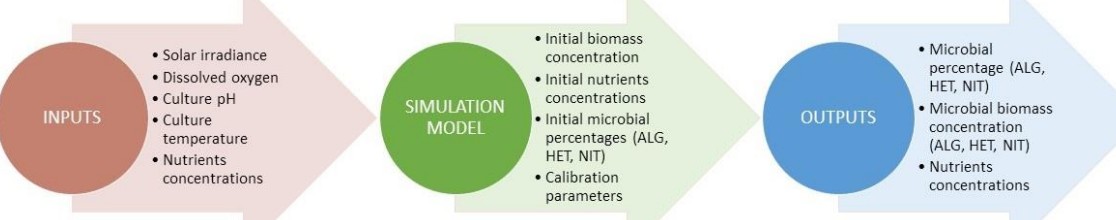

**Figure 3.** Model input-output diagram.

### 3.2. ABACO Model

The biological model has been applied to the treatment of diluted pig slurry as a relevant type of wastewater. The model has been developed considering the main microalgal and bacterial processes that simultaneously occur in the microalgae-based wastewater treatment. An initial dynamic model considering the influence of main environmental variables (irradiance, temperature, pH and dissolved oxygen) on microalgae and bacteria growth was developed by [20]. The model equations were inspired in the BIOALGAE model [17], and it was already validated, the model allowing one to simulate the effect of environmental conditions on the photosynthesis and respiration rate of microalgae–bacteria consortia. Distinctions were performed among activity of microalgae, heterotrophic and nitrifying bacteria [11]. The BIOALGAE model has been improved in this work by considering the influence of nutrients concentration ($CO_2$, $N$-$NH_4^+$, $N$–$NO_3^-$, $P$–$PO_4^{2-}$ and BSOM) in the microalgae and bacteria growth and coefficient yields, resulting in the new ABACO model. The parameters of the model related with the microalgae activity were determined experimentally [21], while the bacterial parameters were obtained from the Activated Sludge Models (ASM) [19,24].

#### 3.2.1. Microalgae Biomass

The microalgal cells are present in the photobioreactors, it not being feed to the system with the influent wastewater. Part of the microalgae biomass is removed every day with the effluent as a function of imposed dilution rate (inverse of hydraulic retention time). Microalgae biomass concentration increases due to autotrophic growth of microalgae, using light as energy source and of $CO_2$ as carbon source, whereas it reduces by endogenous respiration and decay of microalgae. These last phenomena represent the autoxidation of microalgae, where they metabolize their own cellular material. The global balance to estimate the microalgae biomass concentration is given by Equation (2).

$$V \cdot X_{ALG} \cdot \mu_{ALG} = Q_h \cdot X_{ALG} + V \cdot \frac{dX_{ALG}}{dt} \tag{2}$$

where V [$m^3$] is the volume in the reactor, $X_{ALG}$ [$g\ m^{-3}$] is the microalgae biomass concentration, $\mu_{ALG}$ [$day^{-1}$] is the microalgae specific growth rate and $Q_h$ [$m^3\ s^{-1}$] represents the harvesting flow rate.

The specific growth rate $\mu_{ALG}$ is mainly a function of light availability inside the reactor, summarized by the average irradiance inside the culture $I_{av}$ [25], and modified by the influence of different variables such as temperature ($\overline{\mu_{ALG}}(T)$), pH ($\overline{\mu_{ALG}}(pH)$), dissolved oxygen ($\overline{\mu_{ALG}}(DO_2)$) and $CO_2$ ($\overline{\mu_{ALG}}(CO_2)$). In addition the influence of nutrients availability such as ammonium nitrogen ($\overline{\mu_{ALG}}([N - NH_4])$), phosphate phosphorus ($\overline{\mu_{ALG}}([P - PO_4])$) and nitrate nitrogen ($\overline{\mu_{ALG}}([N - NO_3])$), and the microalgae maintenance ($m_{ALG}$), is considered as shown in Equation (3).

$$\mu_{ALG} = (\mu_{ALG}(I_{av}) \cdot \overline{\mu_{ALG}}(T) \cdot \overline{\mu_{ALG}}(pH) \cdot \overline{\mu_{ALG}}(DO_2) \cdot \overline{\mu_{ALG}}(CO_2) \cdot \overline{\mu_{ALG}}(N) \cdot \overline{\mu_{ALG}}([P - PO_4])) - m_{ALG} \tag{3}$$

Microalgae can growth using both ammonium and nitrate as a nitrogen source. Then, there is a process rate for the growth of microalgae using ammonium and another one when using nitrate, thus Equation (3) becomes as Equations (4) and (5) for considering this phenomenon. Notice that Equation (5) considering the consumption of nitrate is only used when there is not ammonium in the system.

$$\mu_{ALG} = (\mu_{ALG}(I_{av}) \cdot \overline{\mu_{ALG}}(T) \cdot \overline{\mu_{ALG}}(pH) \cdot \overline{\mu_{ALG}}(DO_2) \cdot \overline{\mu_{ALG}}(CO_2) \cdot \overline{\mu_{ALG}}([N-NH_4]) \cdot \overline{\mu_{ALG}}([P-PO_4])) - m_{ALG} \quad (4)$$

$$\mu_{ALG} = (\mu_{ALG}(I_{av}) \cdot \overline{\mu_{ALG}}(T) \cdot \overline{\mu_{ALG}}(pH) \cdot \overline{\mu_{ALG}}(DO_2) \cdot \overline{\mu_{ALG}}(CO_2) \cdot \overline{\mu_{ALG}}([N-NO_3]) \cdot \overline{\mu_{ALG}}([P-PO_4])) - m_{ALG} \quad (5)$$

As observed from Equation (3), during the microalgae growth, it is assumed that two main process occur: the microalgal growth and the microalgal maintenance. The microalgae growth rate is modeled as the product of a maximum growth rate ($\mu_{ALG,max}$) as expressed in Equation (6), algae biomass concentration ($X_{ALG}$) as shown in Equation (7), and switching functions for environmental parameters (irradiance, temperature, pH and dissolved oxygen), carbon dioxide, nitrogen and phosphorous (Equation (9)–(15)). The rate of the microalgae maintenance ($m_{ALG}$) considers the endogenous respiration of the microalgae and the microalgae decay (Equation (8)).

Taking the model described by Molina et al. in [25], the light limitation growth model can be expressed as follows:

$$\mu_{ALG}(I_{av}) = \frac{\mu_{ALG,max} \cdot I_{av}^n}{I_k^n + I_{av}^n} \quad (6)$$

where $\mu_{ALG,max}$ [day$^{-1}$] is the maximum microalgae growth rate, $I_{av}$ [$\mu E\ m^{-2}\ s^{-1}$] is the average irradiance inside de culture it summarizing the light availability inside the reactor, $I_k$ [$\mu E\ m^{-2}\ s^{-1}$] is the irradiance constant (equivalent to irradiance required to achieve half of the maximal growth rate) and n is a form parameter. The average irradiance is expressed as follows:

$$I_{av} = \frac{I_0}{K_a \cdot X_{ALG} \cdot h} \left(1 - e^{-K_a \cdot X_{ALG} \cdot h}\right) \quad (7)$$

where $I_0$ [$\mu E\ m^{-2}\ s^{-1}$] is the irradiance on the reactor surface, $K_a$ [$m^2\ g^{-1}$] is the biomass extinction coefficient and h [m] is the culture depth in the reactor.

The endogenous respiration term can be expressed as follows:

$$m_{ALG} = m_{min,alg} + \frac{m_{max,alg} \cdot I_{av}^{n_{resp}}}{I_{k,resp}^{n_{resp}} + I_{av}^{n_{resp}}} \quad (8)$$

where $m_{min,alg}$ and $m_{max,alg}$ [day$^{-1}$] represent the minimum and maximum respiration rates, $I_{k,resp}$ [$\mu E\ m^{-2}\ s^{-1}$] is the irradiance required to stop photosynthesis and start respiration process, and $n_{resp}$ is the form parameter for respiration.

The influence of temperature, pH, dissolved oxygen and nutrients concentration into the microalgae growth rate are included as normalized values, then it varying between 0 and 1. Therefore, when the culture conditions are optimal these terms are equal to 1 and the specific growth rate is only a function of light availability, achieving the maximal value at irradiances upper than saturation irradiance. However, if culture conditions are not optimal the respective normalized values are lower than 1, directly reducing the microalgae growth rate whatever the irradiance. The temperature index $\overline{\mu_{ALG}}(T)$, expressed by Bernard et al. in [26], represents the influence of temperature on microalgae growth. The temperature index can be expressed as follows:

$$\overline{\mu_{ALG}}(T) = \frac{(T - T_{max,\ ALG})(T - T_{min,\ ALG})^2}{(T_{opt,\ ALG} - T_{min,\ ALG})\left(\left((T_{opt,\ ALG} - T_{min,\ ALG})(T - T_{opt,\ ALG})\right) - \left((T_{opt,\ ALG} - T_{max,\ ALG})(T_{opt,\ ALG} + T_{min,\ ALG} - 2 \cdot T)\right)\right)} \quad (9)$$

where T [°C] is the culture temperature, whereas $T_{max}$ [°C], $T_{min}$ [°C] and $T_{opt}$ [°C] are the respective maximal, minimal and optimal temperature for the microalgae strain. As for the

temperature term, the pH term $\overline{\mu_{ALG}}(pH)$ represents the influence of pH on microalgae growth. It can be expressed by a cardinal formula as follows:

$$\overline{\mu_{ALG}}(pH) = \frac{(pH - pH_{max,\ ALG})(pH - pH_{min,\ ALG})^2}{\left(pH_{opt,\ ALG} - pH_{min,\ ALG}\right)\left(\left(\left(pH_{opt,\ ALG} - pH_{min,\ ALG}\right)\left(pH - pH_{opt,\ ALG}\right)\right) - \left(\left(pH_{opt,\ ALG} - pH_{max,\ ALG}\right)\left(pH_{opt,\ ALG} + pH_{min,\ ALG} - 2 \cdot pH\right)\right)\right)} \tag{10}$$

where pH is the culture pH, whereas $pH_{max}$, $pH_{min}$ and $pH_{opt}$ the respective maximal, minimal and optimal pH for the microalgae strain.

The dissolved oxygen term $\overline{\mu_{ALG}}(DO_2)$ depends on a maximum value, determined by the strain, which represents the dissolved oxygen concentration that can be accumulated in the culture without being detrimental to microalgae growth. It can be expressed as the following equation:

$$\overline{\mu_{ALG}}(DO_2) = 1 - \left(\frac{DO_2}{DO_{2,max}}\right)^m \tag{11}$$

where $DO_2$ [%] is the culture dissolved oxygen, $DO_{2,max}$ [%] is the maximum amount of dissolved oxygen for the microalgae strain, m is a form parameter.

The concentration of nutrients in the culture medium (wastewater) can be also a limiting factor for microalgae growth. The influence of carbon dioxide $\overline{\mu_{ALG}}(CO_2)$ is described as follows:

$$\overline{\mu_{ALG}}(CO_2) = \frac{X_{CO_2} + X_{HCO_3}}{K_{S,C,ALG} + X_{CO_2} + X_{HCO_3} + \frac{X_{CO_2}^{n_{C,ALG}}}{K_{I,C,ALG}}} \tag{12}$$

where $X_{CO_2}$ [g m$^{-3}$] is the carbon dioxide concentration, $X_{HCO_3}$ [g m$^{-3}$] is the bicarbonate concentration, $K_{S,C,ALG}$ [g m$^{-3}$] is the microalgae half-saturation constant for carbon, $K_{I,C,ALG}$ [g m$^{-3}$] is the microalgae inhibition constant for carbon, and $n_{C,ALG}$ is the microalgae form parameter for carbon. The influence of ammonium nitrogen $\overline{\mu_{ALG}}([N - NH_4])$ is represented by the following equation:

$$\overline{\mu_{ALG}}([N - NH_4]) = \frac{X_{NH_4}}{X_{NH_4} + K_{S,NH_4,ALG} + \frac{X_{NH_4}^{n_{NH_4,ALG}}}{K_{I,NH_4,ALG}}} \tag{13}$$

where $X_{NH_4}$ [g m$^{-3}$] is the ammonium nitrogen concentration, $K_{S,NH_4,ALG}$ [g m$^{-3}$] is the microalgae half-saturation constant for ammonium, $K_{I,NH_4,ALG}$ [g m$^{-3}$] is the microalgae inhibition constant for ammonium, and $n_{NH_4,ALG}$ is the microalgae form parameter for ammonium. The influence of nitrate nitrogen $\overline{\mu_{ALG}}([N - NO_3])$.is represented by the following equation:

$$\overline{\mu_{ALG}}([N - NO_3]) = \frac{X_{NO_3}}{X_{NO_3} + K_{S,NO_3,ALG} + \frac{X_{NO_3}^{n_{NO_3,ALG}}}{K_{I,NO_3,ALG}}} \tag{14}$$

where $X_{NO_3}$ [g m$^{-3}$] is the nitrate nitrogen concentration, $K_{S,NO_3,ALG}$ [g m$^{-3}$] is the microalgae half-saturation constant for nitrate, $K_{I,NH_4,ALG}$ [g m$^{-3}$] is the microalgae inhibition constant for nitrate, and $n_{NO_3,ALG}$ is the microalgae form parameter for nitrate. The influence of phosphate phosphorus $\overline{\mu_{ALG}}([P - PO_4])$ is represented by the following equation:

$$\overline{\mu_{ALG}}([P - PO_4]) = \frac{X_{PO_4}}{X_{PO_4} + K_{S,PO_4,ALG}} \tag{15}$$

where $X_{PO_4}$ [g m$^{-3}$] is the phosphate phosphorus concentration and $K_{S,PO_4,ALG}$ [g m$^{-3}$] is the microalgae half-saturation constant for phosphate.

### 3.2.2. Heterotrophic Bacteria

Heterotrophic bacteria are already present in the influent wastewater, then they are supplied to the system with the inlet wastewater also it being removed with the harvested flow rate as a function of imposed dilution rate. Heterotrophic bacteria grow using the organic matter as source of energy and carbon. These bacteria are aerobic then consuming $O_2$ produced during the photosynthesis process. The endogenous respiration and the decay are responsible for the heterotrophic biomass lost. The global balance to estimate the heterotrophic bacteria concentration is given by Equation (16).

$$Q_d \cdot X_{HET,in} + V \cdot X_{HET,out} \cdot \mu_{HET} = Q_h \cdot X_{HET,out} + V \cdot \frac{dX_{HET,out}}{dt} \qquad (16)$$

where $Q_d$ [m$^3$ s$^{-1}$] is the dilution flow rate, $X_{HET,in}$ [g m$^{-3}$] is the heterotrophic bacteria inlet concentration, $\mu_{HET}$ [day$^{-1}$] is the specific growth rate of heterotrophic bacteria and $X_{HET,out}$ [g m$^{-3}$] is the heterotrophic bacteria concentration in the reactor.

As with microalgal processes, heterotrophic processes include both the heterotrophic growth and the heterotrophic maintenance. The heterotrophic specific growth rate $\mu_{HET}$ is modeled as the product of maximum growth rate ($\mu_{HET,max}$) and switching functions for environmental parameters such as temperature ($\overline{\mu_{HET}}(T)$), pH ($\overline{\mu_{HET}}(pH)$) and dissolved oxygen ($\overline{\mu_{HET}}(DO_2)$); in addition to biodegradable soluble organic matter ($\overline{\mu_{HET}}(BSOM)$), ammonium nitrogen ($\overline{\mu_{HET}}([N-NH_4])$) and phosphate phosphorous ($\overline{\mu_{HET}}([P-PO_4])$) (Equation (17)). The rate of the heterotrophic maintenance ($m_{HET}$) considers the endogenous respiration of the heterotrophic bacteria and the heterotrophic decay. The specific growth rate for heterotrophic bacteria is expressed from the following equation:

$$\mu_{HET} = \mu_{HET,max} \cdot (\overline{\mu_{HET}}(T) \cdot \overline{\mu_{HET}}(pH) \cdot \overline{\mu_{HET}}(DO_2) \cdot \overline{\mu_{HET}}([N-NH_4]) \cdot \overline{\mu_{HET}}([P-PO_4]) \cdot \overline{\mu_{HET}}(BSOM)) - m_{HET} \qquad (17)$$

where $\mu_{HET,max}$ [day$^{-1}$] is the maximum specific growth rate for heterotrophic bacteria, whereas $m_{HET}$ [day$^{-1}$] represent the endogenous respiration of the heterotrophic bacteria and the heterotrophic decay.

The temperature and pH terms ($\overline{\mu_{HET}}(T)$ and $\overline{\mu_{HET}}(pH)$) are also based on the cardinal model, so they are identical to those previously expressed for microalgae. These terms depend on the maximum ($T_{max,\ HET}$ and $pH_{max,\ HET}$), minimum ($T_{min,\ HET}$ and $pH_{min,\ HET}$) and optimal ($T_{opt,\ HET}$ and $pH_{opt,\ HET}$) values of temperature and pH for heterotrophic bacteria, such as expressed in Equations (18) and (19), respectively.

$$\overline{\mu_{HET}}(T) = \frac{(T-T_{max,\ HET})(T-T_{min,\ HET})^2}{(T_{opt,\ HET}-T_{min,\ HET})\left(\left(\left(T_{opt,\ HET}-T_{min,\ HET}\right)\left(T-T_{opt,\ HET}\right)\right)-\left(\left(T_{opt,\ HET}-T_{max,\ HET}\right)\left(T_{opt,\ HET}+T_{min,\ HET}-2\cdot T\right)\right)\right)} \qquad (18)$$

$$\overline{\mu_{HET}}(pH) = \frac{(pH-pH_{max,\ HET})(pH-pH_{min,\ HET})^2}{\left(pH_{opt,\ HET}-pH_{min,\ HET}\right)\left(\left(\left(pH_{opt,\ HET}-pH_{min,\ HET}\right)\left(pH-pH_{opt,\ HET}\right)\right)-\left(\left(pH_{opt,\ HET}-pH_{max,\ HET}\right)\left(pH_{opt,\ HET}+pH_{min,\ HET}-2\cdot pH\right)\right)\right)} \qquad (19)$$

The influence of dissolved oxygen $\overline{\mu_{HET}}(DO_2)$ is expressed as follows:

$$\overline{\mu_{HET}}(DO_2) = \frac{DO_2}{DO_2 + K_{S,DO_2,\ HET}} \qquad (20)$$

where $K_{S,DO_2,\ HET}$ [g m$^{-3}$] is the heterotrophic bacteria half-saturation constant for dissolved oxygen. The influence of ammonium nitrogen $\overline{\mu_{HET}}([N-NH_4])$ is represented by the following equation:

$$\overline{\mu_{HET}}([N-NH_4]) = \frac{X_{NH_4}}{X_{NH_4} + K_{S,NH_4,HET}} \qquad (21)$$

where $K_{S,NH_4,HET}$ [g m$^{-3}$] is the heterotrophic bacteria half-saturation constant for ammonium. The influence of phosphate phosphorus $\overline{\mu_{HET}}([P - PO_4])$ is represented by the following equation:

$$\overline{\mu_{HET}}([P - PO_4]) = \frac{X_{PO_4}}{X_{PO_4} + K_{S,PO_4,HET}} \tag{22}$$

where $K_{S,PO_4,HET}$ [g m$^{-3}$] is the heterotrophic bacteria half-saturation constant for phosphate. The influence of biodegradable soluble organic matter $\overline{\mu_{HET}}(BSOM)$ is represented by the following equation:

$$\overline{\mu_{HET}}(BSOM) = \frac{X_{BSOM}}{X_{BSOM} + K_{S,BSOM,HET}} \tag{23}$$

where $X_{BSOM}$ [g m$^{-3}$] is the concentration of biodegradable soluble organic matter (BSOM) in the reactor and $K_{S,BSOM,HET}$ [g m$^{-3}$] is the heterotrophic bacteria half-saturation constant for BSOM.

### 3.2.3. Nitrifying Bacteria

Nitrifiying bacteria can be also supplied to the reactor with the wastewater supplied to the reactor, it being also removed during harvesting. Nitrifying bacteria perform the nitrification process, thus oxidizing ammonium to nitrate. These microorganisms are aerobic, then requiring oxygen, also using $CO_2$ as a carbon source. The concentration of nitrifying bacteria increases due to growth but also decrease by endogenous respiration and decay. The global balance to estimate the concentration of nitrifying bacteria is given by Equation (24).

$$Q_d \cdot X_{NIT,in} + V \cdot X_{NIT,out} \cdot \mu_{NIT} = Q_h \cdot X_{NIT,out} + V \cdot \frac{dX_{NIT,out}}{dt} \tag{24}$$

where $X_{NIT,in}$ [g m$^{-3}$] is the nitrifying bacteria inlet concentration, $\mu_{NIT}$ [day$^{-1}$] is the nitrifying bacteria specific growth rate and $X_{NIT,out}$ [g m$^{-3}$] is the nitrifying bacteria concentration in the reactor.

The processes related with nitrifying bacteria include both autotrophic growth and maintenance. The rate of the autotrophic growth is modeled as the product of maximum growth rate ($\mu_{NIT,max}$) and switching functions for environmental parameters, such as temperature ($\overline{\mu_{HET}}(T)$), pH ($\overline{\mu_{HET}}(pH)$) and dissolved oxygen ($\overline{\mu_{HET}}(DO_2)$); in addition to ammonium nitrogen ($\overline{\mu_{HET}}([N - NH_4])$) and phosphate phosphorous ($\overline{\mu_{HET}}([P - PO_4])$) (Equation (25)). The rate of maintenance ($m_{NIT}$) considers the endogenous respiration of the nitrifying bacteria and nitrifying decay. The following equation represents the nitrifying bacteria specific growth rate:

$$\mu_{NIT} = \mu_{NIT,max} \cdot (\overline{\mu_{NIT}}(T) \cdot \overline{\mu_{NIT}}(pH) \cdot \overline{\mu_{NIT}}(DO_2) \cdot \overline{\mu_{NIT}}(CO_2) \cdot \overline{\mu_{NIT}}([N - NH_4]) \cdot \overline{\mu_{NIT}}([P - PO_4])) - m_{NIT} \tag{25}$$

where $\mu_{NIT,max}$ [day$^{-1}$] is the maximum specific growth rate for nitrifying bacteria and $m_{NIT}$ [day$^{-1}$] is the endogenous respiration of the nitrifying bacteria and the nitrifying maintenance.

As for the heterotrophic bacteria, the temperature and pH terms ($\overline{\mu_{NIT}}(T)$ and $\overline{\mu_{NIT}}(pH)$) are expressed the same form. These terms depend on the maximum ($T_{max,\ NIT}$ and $pH_{max,\ NIT}$), minimum ($T_{min,\ NIT}$ and $pH_{min,\ NIT}$) and optimal ($T_{opt,\ NIT}$ and $pH_{opt,\ NIT}$) values of temperature and pH for nitrifying bacteria, such as expressed in Equations (26) and (27), respectively.

$$\overline{\mu_{NIT}}(T) = \frac{(T - T_{max,\ NIT})(T - T_{min,\ NIT})^2}{(T_{opt,\ NIT} - T_{min,\ NIT})\left(\left(\left(T_{opt,\ NIT} - T_{min,\ NIT}\right)\left(T - T_{opt,\ NIT}\right)\right) - \left(\left(T_{opt,\ NIT} - T_{max,\ NIT}\right)\left(T_{opt,\ NIT} + T_{min,\ NIT} - 2 \cdot T\right)\right)\right)} \tag{26}$$

$$\overline{\mu_{NIT}}(pH) = \frac{(pH - pH_{max,\ NIT})(pH - pH_{min,\ NIT})^2}{(pH_{opt,\ NIT} - pH_{min,\ NIT})\left(\left(\left(pH_{opt,\ NIT} - pH_{min,\ NIT}\right)\left(pH - pH_{opt,\ NIT}\right)\right) - \left(\left(pH_{opt,\ NIT} - pH_{max,\ NIT}\right)\left(pH_{opt,\ NIT} + pH_{min,\ NIT} - 2 \cdot pH\right)\right)\right)} \tag{27}$$

The influence of dissolved oxygen $\overline{\mu_{NIT}}(DO_2)$ is represented by the following equation:

$$\overline{\mu_{NIT}}(DO_2) = \frac{DO_2}{\left(DO_2 + K_{S,DO_2, NIT}\right) \cdot \left(1 + \frac{DO_2}{K_{I,DO_2, NIT}}\right)} \tag{28}$$

where $K_{S,DO_2, NIT}$ [g m$^{-3}$] is the nitrifying bacteria half-saturation constant for dissolved oxygen and $K_{I,DO_2, NIT}$ [g m$^{-3}$] is the nitrifying bacteria inhibition constant for dissolved oxygen.

The influence of carbon dioxide $\overline{\mu_{NIT}}(CO_2)$ is described as follows:

$$\overline{\mu_{NIT}}(CO_2) = \frac{X_{CO_2} + X_{HCO_3}}{K_{S,C,NIT} + X_{CO_2} + X_{HCO_3}} \tag{29}$$

where $K_{S,C,NIT}$ [g m$^{-3}$] is the nitrifying bacteria half-saturation constant for carbon.

The influence of ammonium nitrogen $\overline{\mu_{NIT}}([N - NH_4])$ is represented by the following equation:

$$\overline{\mu_{NIT}}([N - NH_4]) = \frac{X_{NH_4}}{X_{NH_4} + K_{S,NH_4,NIT}} \tag{30}$$

where $K_{S,NH_4,NIT}$ [g m$^{-3}$] is the nitrifying bacteria half-saturation constant for ammonium.

The influence of phosphate phosphorus $\overline{\mu_{NIT}}([P - PO_4])$ is represented by the following equation:

$$\overline{\mu_{NIT}}([P - PO_4]) = \frac{X_{PO_4}}{X_{PO_4} + K_{S,PO_4,NIT}} \tag{31}$$

where $K_{S,PO_4,NIT}$ [g m$^{-3}$] is the nitrifying bacteria half-saturation constant for phosphate.

### 3.2.4. Dissolved Oxygen

During the photosynthesis, microalgae release $O_2$ and its consumed by aerobic bacteria respiration and microalgae respiration. The dissolved oxygen is measured and represents a model input.

### 3.2.5. Dissolved Carbon Dioxide

Carbon dioxide is generated during the aerobic respiration (bacteria and microalgae), and is consumed by nitrifying bacteria as carbon source and by microalgae for the photosynthetic process. The concentration of $CO_2$ is determined by the total inorganic carbon concentration and the presence of bicarbonate buffer. Thus, it is assumed that $CO_2$ is always in chemical equilibrium with bicarbonate ($HCO_3$) and carbonate ($CO_3$). The following equilibrium constant between carbon dioxide, carbonate and bicarbonate is defined:

$$K_1 = \frac{[X_{HCO_3}][H^+]}{[X_{CO_2}]} = 10^{-6.381} \tag{32}$$

$$K_2 = \frac{[X_{CO_3}][H^+]}{[X_{HCO_3}]} = 10^{-10.377} \tag{33}$$

where $X_{HCO_3}$ is the bicarbonate concentration, $X_{CO_2}$ is the carbon dioxide concentration, $X_{CO_3}$ is the carbonate concentration, and $H^+$ is the concentration of hydrogen ions, which can be obtained from the pH by means of the following equation:

$$H^+ = 10^{-pH} \tag{34}$$

Assuming a total inorganic carbon concentration $X_{C_T}$ of 0.1 [g L$^{-1}$], the concentration of bicarbonate and carbon dioxide can be obtained from the following equations:

$$X_{HCO_3} = \frac{\left(H^+ \cdot X_{C_T}\right)}{\left(K_2 + H^+ + H^{+2}\right)} \tag{35}$$

$$X_{CO_2} = \frac{\left(X_{HCO_3} \cdot H^+\right)}{K_1} \tag{36}$$

### 3.2.6. Chemical Oxygen Demand

Chemical oxygen demand (COD) of the inlet wastewater is mainly related with the organic matter already present on it. The COD includes the total organic matter, both the biodegradable and the no biodegradable organic matter. Additionally, it is produced during the microbial decay, and the biodegradable fraction is consumed by heterotrophic bacteria.

### 3.2.7. Biodegradable Organic Soluble Matter.

The biodegradable organic matter dissolved is the fraction of the organic matter which is available for biodegradation by heterotrophic bacteria $X_{HET}$. It is introduced in the influent wastewater and is produced by microbial decay. $X_{BSOM}$ is removed by heterotrophic consumption and during the dilution process, such as expressed in the following equations:

$$
\begin{aligned}
Q_d \cdot X_{BSOM,in} + V \cdot & \\
\cdot \left( X_{ALG} \cdot \mu_{alg} \cdot Y_{gen}\left[\frac{BSOM}{alg}\right] + X_{het,out} \cdot \mu_{het} \cdot Y_{gen}\left[\frac{BSOM}{het}\right] + X_{nit,out} \cdot \mu_{nit} \cdot Y_{gen}\left[\frac{BSOM}{nit}\right] \right) &= \\
= Q_h \cdot X_{BSOM,out} + V \cdot & \\
\cdot \left( X_{het,out} \cdot \mu_{het} \cdot Y_{con}\left[\frac{PO_4}{het}\right] \right) + V \cdot \frac{dX_{BSOM,out}}{dt} &
\end{aligned}
\tag{37}
$$

where $X_{BSOM,in}$ [g m$^{-3}$] is the inlet BSOM concentration, $X_{BSOM,out}$ [g m$^{-3}$] the BSOM concentration in the reactor, $Y_{gen}\left[\frac{BSOM}{alg}\right]$ [-] represents the BSOM generation rate from microalgae, $Y_{gen}\left[\frac{BSOM}{het}\right]$ [-] is the BSOM generation rate from heterotrophic bacteria, $Y_{gen}\left[\frac{BSOM}{nit}\right]$ [-] is the BSOM generation rate from nitrifying bacteria, and $Y_{con}\left[\frac{PO_4}{het}\right]$ [-] shows the BSOM consumption rate from heterotrophic bacteria.

### 3.2.8. Ammonium Nitrogen

Different forms of nitrogen can be found in wastewater. Ammonium nitrogen is introduced in the system though the influent wastewater, it being consumed by microalgae, heterotrophic bacteria, and nitrifying bacteria. Besides, ammonium nitrogen is generated by microbial decay. The ammonium nitrogen concentration is modelled by the following equation:

$$
\begin{aligned}
Q_d \cdot X_{NH_4,in} = Q_h \cdot X_{NH_4,out} + V \cdot & \\
\cdot \left( X_{ALG} \cdot \mu_{alg} \cdot Y_{con}\left[\frac{NH_4}{alg}\right] + X_{het,out} \cdot \mu_{het} \cdot Y_{con}\left[\frac{NH_4}{het}\right] + X_{nit,out} \cdot \mu_{nit} \cdot Y_{con}\left[\frac{NH_4}{nit}\right] \right) + V \cdot \frac{dX_{NH_4,out}}{dt} &
\end{aligned}
\tag{38}
$$

where $X_{NH_4,in}$ [g m$^{-3}$] is the ammonium nitrogen inlet concentration, $X_{NH_4,out}$ [g m$^{-3}$] represents the outlet ammonium nitrogen concentration, $Y_{con}\left[\frac{NH_4}{alg}\right]$ [-] shows the ammonium consumption rate from microalgae, $Y_{con}\left[\frac{NH_4}{het}\right]$ [-] is the ammonium consumption rate from heterotrophic bacteria and $Y_{con}\left[\frac{NH_4}{nit}\right]$ [-] shows the ammonium consumption rate from nitrifying bacteria.

### 3.2.9. Nitrate Nitrogen

Nitrogen in form of nitrate enters in the system through the influent wastewater and it is produced during nitrification by nitrifying bacteria. It is consumed by microalgae

cells when ammonium is not presented or have been consumed. The nitrate nitrogen concentration is modelled by the following equation:

$$Q_d \cdot X_{NO_3,in} + V \cdot X_{NO_3,out} \cdot \mu_{nit} \cdot Y_{gen}\left[\frac{NO_3}{nit}\right] = Q_h \cdot X_{NO_3,out} + V \cdot$$
$$\cdot \left(X_{ALG} \cdot \mu_{alg} \cdot Y_{con}\left[\frac{NO_3}{alg}\right]\right) + V \cdot \frac{dX_{NO_3,out}}{dt} \tag{39}$$

where $X_{NO_3,in}$ [g m$^{-3}$] is the inlet nitrate nitrogen concentration, $X_{NO_3,out}$ [g m$^{-3}$] represents the outlet nitrate nitrogen concentration, $Y_{gen}\left[\frac{NO_3}{nit}\right]$ [-] is the nitrate generation rate from nitrifying bacteria and $Y_{con}\left[\frac{NO_3}{alg}\right]$ [-] shows the nitrate consumption rate from microalgae.

### 3.2.10. Phosphate Phosphorous

Phosphorous is contained into the wastewater both as organic and inorganic. Organic phosphorous is transformed into inorganic during degradation of biodegradable organic matter then the phosphate phosphorous concentration corresponding to total phosphorous available. Phosphate phosphorous is introduced in the system with influent wastewater, it being produced during decay of all microbial populations. It is consumed during the growth of microalgae, heterotrophic bacteria and nitrifying bacteria. The phosphate phosphorus concentration is modelled by the following equation:

$$Q_d \cdot X_{PO_4,in} = Q_h \cdot X_{PO_4,out} + V \cdot$$
$$\cdot \left(X_{ALG} \cdot \mu_{alg} \cdot Y_{con}\left[\frac{PO_4}{alg}\right] + X_{het,out} \cdot \mu_{het} \cdot Y_{con}\left[\frac{PO_4}{het}\right] + X_{nit,out} \cdot \mu_{nit} \cdot Y_{con}\left[\frac{PO_4}{nit}\right]\right) + V \cdot \frac{dX_{PO_4,out}}{dt} \tag{40}$$

where $X_{PO_4,in}$ [g m$^{-3}$] is the inlet phosphate phosphorus concentration, $X_{PO_4,out}$ [g m$^{-3}$] is the outlet phosphate phosphorus concentration, $Y_{con}\left[\frac{PO_4}{alg}\right]$ [-] represents the phosphate consumption rate from microalgae, $Y_{con}\left[\frac{PO_4}{het}\right]$ [-] is the phosphate consumption rate from heterotrophic bacteria and $Y_{con}\left[\frac{PO_4}{nit}\right]$ [-] is the phosphate consumption rate from nitrifying bacteria.

Although growth rate models for the different microorganisms are well defined, the consumption and generation parameters of nutrients associated with each species present some uncertainty. The production of microalgae using wastewater as culture medium presents diverse variability in the model parameters. Depending on the type of wastewater and its components, the generation and consumption parameters associated with microalgae and bacteria may vary. This fact raises the need for a model that allows adapting its parameters for each situation. Therefore, a calibration method is presented using genetic algorithms that is capable of estimating the characteristic parameters of the model from experimental data measured in the reactor.

### 3.3. Experimental Datasets

Experimental data for model calibration and validation were collected from two laboratory-scale photobioreactors, which were fed with pig slurry diluted at 20%. The concentrations of biomass and the major nutrients (N–NH$_4^+$, N–NO$_3^-$, P–PO$_4^{2-}$, COD) both at the inlet wastewater and inside the reactor were measured. The descriptions of the reactors and the probes used to collect the data (temperature, pH, DO, and light), along with the methods used to measure biomass and nutrients, are shown in Section 2.

### 3.4. Calibration Process

The already shown equations of the model include of a series of characteristic parameters whose exact values are unknown, or the values are known in a defined range. The uncertainty in the values of these parameters imposes the need for a calibration process, which has been carried out through genetic algorithms. Calibration using genetic algorithms results in a useful and reliable method for the estimation of uncertain parameters, since it allows optimizing a cost function that measures the deviation of the output of the model from that of the real system by modifying the parameter values between the

established limits. The ranges of the estimated parameters have been obtained from the cited literature, and from experience in the design of the installation.

The calibration process using genetic algorithms was implemented in MATLAB using the Genetic Algorithm Optimization Toolbox (GAOT), based on [1], with an initial population of 50 phenotypes (solutions) and a termination condition of 50 generations. This method starts with an initial set of calibration parameters and runs the model to obtain the error. The cost function is computed as the sum of the individual root mean square error (RMSE) functions for the simulated organism and nutrients (total biomass, ammonium, nitrate, phosphate, and BSOM) and the real measured values, expressed as the following equation:

$$J = \left( \sqrt{\sum_{i=1}^{N} \frac{\left(Cb_{total_{est}}(i) - Cb_{total_{real}}(i)\right)^2}{N}} \right) + \left( \sqrt{\sum_{i=1}^{N} \frac{\left(X_{NH_4,est}(i) - X_{NH_4,real}(i)\right)^2}{N}} \right) + \left( \sqrt{\sum_{i=1}^{N} \frac{\left(X_{NO_3,est}(i) - X_{NO_3,real}(i)\right)^2}{N}} \right)$$

$$+ \left( \sqrt{\sum_{i=1}^{N} \frac{\left(X_{PO_4,est}(i) - X_{PO_4,real}(i)\right)^2}{N}} \right) + \left( \sqrt{\sum_{i=1}^{N} \frac{\left(X_{BSOM_{est}}(i) - X_{BSOM_{real}}(i)\right)^2}{N}} \right)$$

where $Cb_{total_{est}}$ $[g\,m^{-3}]$ is the estimated total biomass concentration (microalgae + heterotrophic bacteria + nitrifying bacteria); $Cb_{total_{real}}$ $[g\,m^{-3}]$ is the experimental total biomass concentration measured. The rests of the parameters also describe the differences between the estimated concentrations and the experimentally measured ones for all elements. N represents the size of the data vector.

The calibration parameters are related to the maximum growth rates for the microorganisms, and the coefficients of generation and nutrient consumption. Table 2 lists the descriptions of all the calibration parameters, and the values obtained as a result of the calibration process.

In addition to the parameters described in the table, through this calibration process, it is possible to estimate the percentage of each species in the reactor. The experimental measurement of the concentration of the species of bacteria is something complex to carry out and highlights the need for a simple way of being able to estimate the percentage of each species within the reactor. Therefore, for both the calibration and validation data, the genetic algorithm method was used to determine the initial percentage of each species. In this way, the calibration process acts as a tool to estimate the percentages of microalgae and bacteria involved in the reactor from the measurements of total biomass and nutrients in it.

Data used during the calibration process correspond to the experimental measurements from during for 14 consecutive days. The imposed culture conditions were equivalent to that found in a raceway reactor, with light and dark cycles representing day and night. In addition, pH and dissolved oxygen were controlled by injecting $CO_2$ and air. Figure 4 represents the experimental data measured, which correspond to measurements of irradiance, pH, dissolved oxygen, and temperature, in addition to measurements of total biomass dry weight (microalgae, heterotrophic bacteria, and nitrifying bacteria) and measurements of nutrients (ammonium, nitrate, phosphate, and BSOM).

Figure 5 represents the calibration results obtained in the estimation of the model variables. This figure is made up of six independent graphs that represent different variables estimated in the model. Figure 5a represents the percentage of each species of microorganisms within the reactor. Figure 5b represents the biomass concentration for each organism in the reactor (microalgae, heterotrophic bacteria, and nitrifying bacteria), in addition to the total biomass concentration, expressed as the sum of the individual concentrations, and the experimental measurements. Figure 5c represents the estimated phosphate concentration and the experimental data. Figure 5d shows the estimated ammonium concentration and the experimental values. Figure 5e represents the estimated nitrate concentration and the experimental measurements. Finally, Figure 5f represents the estimated biodegradable soluble organic matter concentration, compared with the experimental measurement.

As a result of the calibration, initial percentages of 82.1% for microalgae, 13.2% for heterotrophic bacteria, and 4.7% for nitrifying bacteria have been established. Looking at Figure 5a,b, it is observed how the concentration of microalgae decreases until reaching a steady state. On the other hand, the concentration of heterotrophic bacteria grows slightly, consuming ammonium and organic matter, while the concentration of nitrifying bacteria remains constant. The sum of the concentration of each species represents the total biomass concentration (dashed line), which properly fit to the experimental data.

**Table 2.** Calibration parameters for the ABACO model.

| Symbol | Parameter | Value | Unit |
|--------|-----------|-------|------|
| $\mu_{alg,max}$ | Microalgae maximum growth rate | 1.591 | $day^{-1}$ |
| $\mu_{het,max}$ | Heterotrophic bacteria maximum growth rate | 1.235 | $day^{-1}$ |
| $\mu_{nit,max}$ | Nitrifying bacteria maximum growth rate | 0.730 | $day^{-1}$ |
| $m_{min,alg}$ | Microalgae endogenous respiration minimum rate | 0.01 | $day^{-1}$ |
| $m_{max,alg}$ | Microalgae endogenous respiration maximum rate | 0.276 | $day^{-1}$ |
| $Y_{con}\left[\frac{NH_4}{alg}\right]$ | Ammonium consumption rate from microalgae | 0.369 | $g_{NH_4}\,g_{alg}^{-1}$ |
| $Y_{con}\left[\frac{NO_3}{alg}\right]$ | Nitrate consumption rate from microalgae | 0.214 | $g_{NO_3}\,g_{alg}^{-1}$ |
| $Y_{con}\left[\frac{PO_4}{alg}\right]$ | Phosphate consumption rate from microalgae | 0.008 | $g_{PO_4}\,g_{alg}^{-1}$ |
| $Y_{gen}\left[\frac{BSOM}{alg}\right]$ | BSOM generation rate from microalgae | 0.148 | $g_{BSOM}\,g_{alg}^{-1}$ |
| $Y_{con}\left[\frac{NH_4}{het}\right]$ | Ammonium consumption rate from heterotrophic bacteria | 0.299 | $g_{NH_4}\,g_{het}^{-1}$ |
| $Y_{con}\left[\frac{PO_4}{het}\right]$ | Phosphate consumption rate from heterotrophic bacteria | 0.017 | $g_{PO_4}\,g_{het}^{-1}$ |
| $Y_{gen}\left[\frac{BSOM}{het}\right]$ | BSOM generation rate from heterotrophic bacteria | 0.153 | $g_{BSOM}\,g_{het}^{-1}$ |
| $Y_{con}\left[\frac{BSOM}{het}\right]$ | BSOM consumption rate from heterotrophic bacteria | 0.478 | $g_{BSOM}\,g_{het}^{-1}$ |
| $Y_{con}\left[\frac{NH_4}{nit}\right]$ | Ammonium consumption rate from nitrifying bacteria | 3.224 | $g_{NH_4}\,g_{nit}^{-1}$ |
| $Y_{gen}\left[\frac{NO_3}{nit}\right]$ | Nitrate generation rate from nitrifying bacteria | 0.355 | $g_{NO_3}\,g_{nit}^{-1}$ |
| $Y_{con}\left[\frac{PO_4}{nit}\right]$ | Phosphate consumption rate from nitrifying bacteria | 0.182 | $g_{PO_4}\,g_{nit}^{-1}$ |
| $Y_{gen}\left[\frac{BSOM}{nit}\right]$ | BSOM generation rate from nitrifying bacteria | 0.149 | $g_{BSOM}\,g_{nit}^{-1}$ |

Although the experimental concentrations of nutrients (phosphate, ammonium, nitrate, and BSOM) are very scattered, a trend is observed for each. The estimated values for the elements presented in Figure 5c–f fit correctly within the experimental data. The total RMSE value obtained through the cost function during calibration was 25.93, which is an acceptable value, since, with the exception of ammonia, the range of variation of the variables analyzed is small. An error of 0.076 was obtained for total biomass concentration, an error of 0.9 for phosphate, an error of 20.76 for ammonium, an error of 1.27 for nitrate, and an error of 2.94 for BSOM.

### 3.5. Validation

The validation data used to verify the values of the characteristic parameters obtained during the calibration process were obtained in a separate vessel reactor, operated in parallel with the one used for calibration. These data collect the experimental measurements from 14 days, represented in Figure 6.

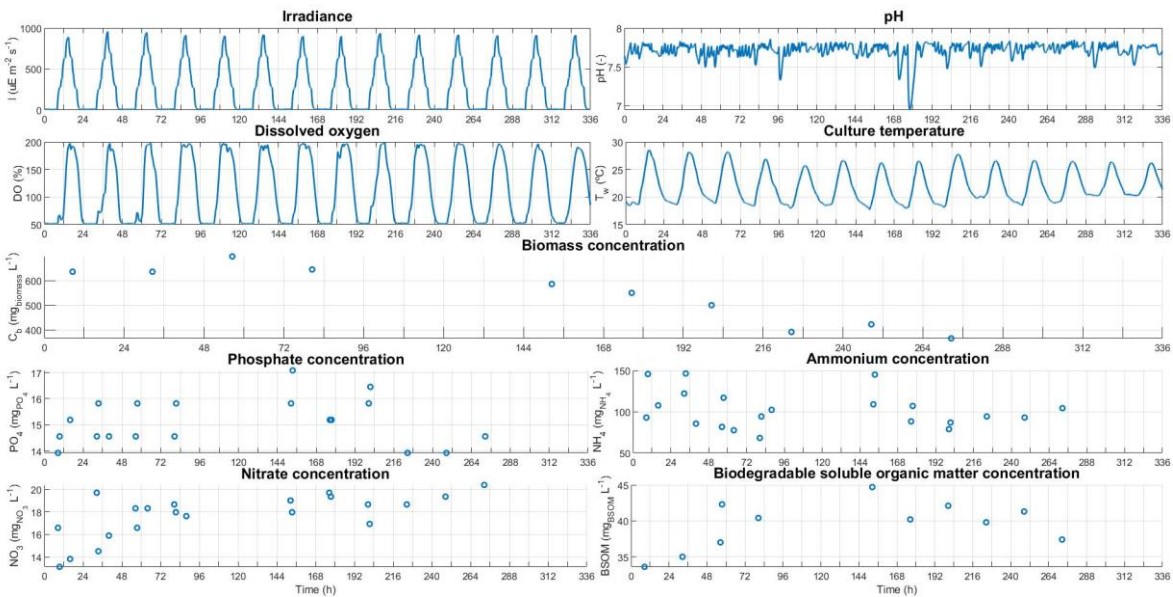

**Figure 4.** Input variables for calibration.

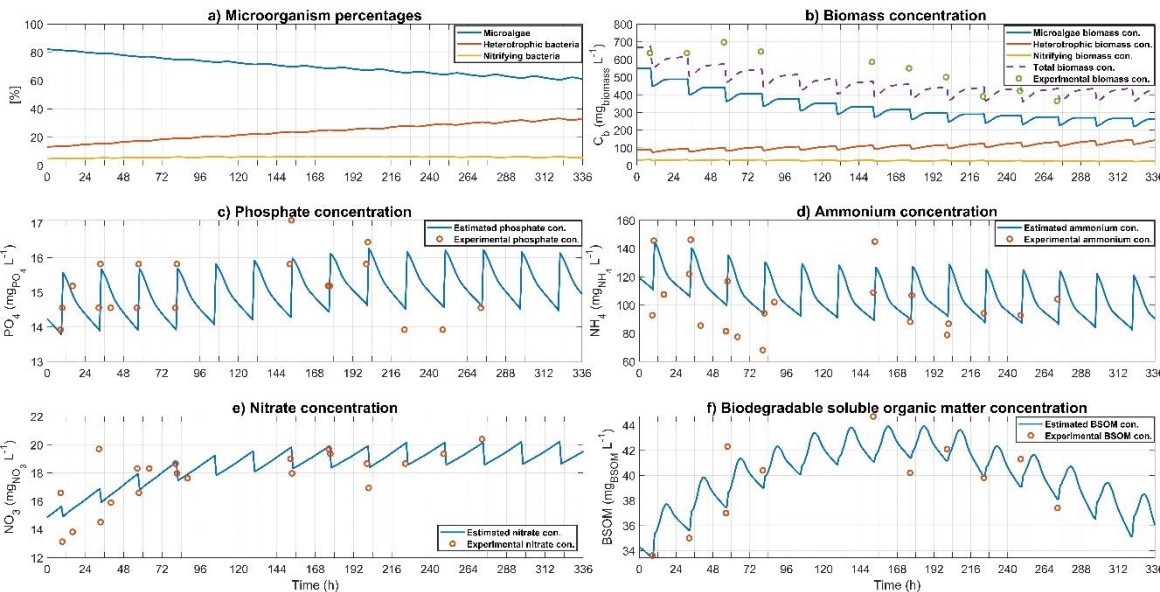

**Figure 5.** Calibration results for the biomass production model with wastewater medium. (**a**) Microalgae and bacteria percentages inside the reactor; (**b**) Microalgae and bacteria biomass concentration; (**c**) Phosphate concentration inside the reactor; (**d**) Ammonium concentration inside the reactor; (**e**) Nitrate concentration inside the reactor; (**f**) Biodegradable soluble organic matter concentration inside the reactor.

For the validation process, calibration using genetic algorithms has been used to determine the initial percentages of microorganisms in the reactor. In this way, it is possible to estimate the starting points for the concentration of microalgae and bacteria. In this case, the initial percentages obtained were 85% for microalgae, 12.6% for heterotrophic bacteria, and 2.4% for nitrifying bacteria, very similar to the percentages obtained during the calibration test. After this initial point, the concentrations of all the elements in the reactor were estimated and compared with the points measured experimentally, represented in Figure 7.

Figure 7a,b shows trends in biomass concentrations similar to those obtained during calibration. The concentration of microalgae decreases till achieving steady state, the

heterotrophic bacteria slightly grow, and the nitrifying bacteria remain constant. The total concentration correctly resembles the trend shown by the experimental measurements.

The estimation of the phosphate concentration (Figure 7c) shows an increasing trend, slightly away from the center of the measurement points. However, the estimation is within the range of the experimental values. The concentration of ammonium (Figure 7d) maintains a good trend within the established range, as does the estimated nitrate concentration (Figure 7e). Finally, the BSOM estimation (Figure 7f) shows a trend similar to the calibration results, within the experimental points. In this case, the total RMSE error was 30.25, slightly higher for calibration.

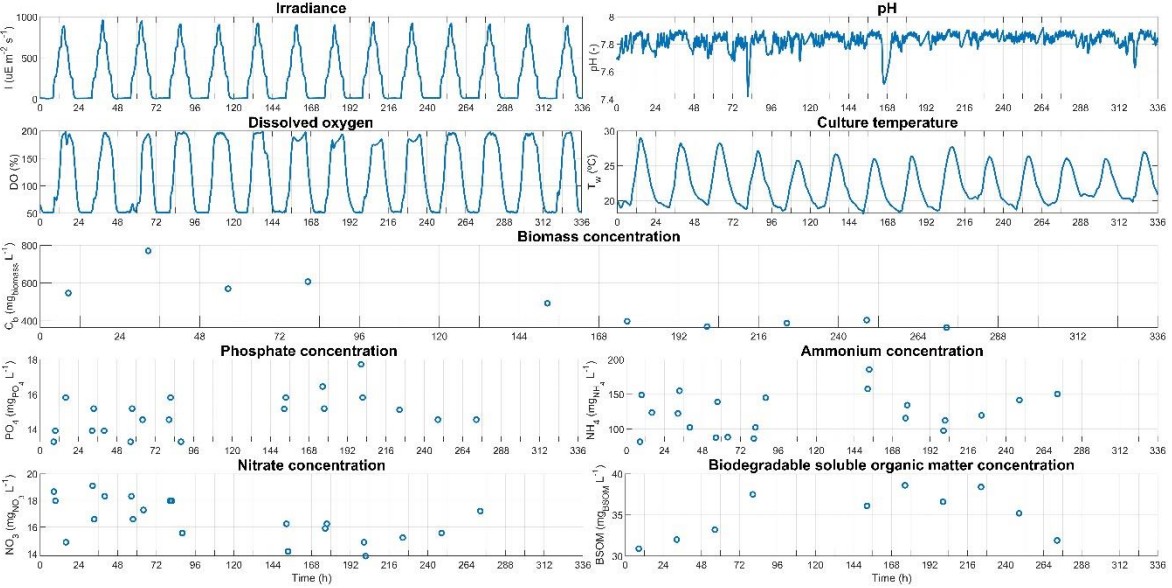

**Figure 6.** Input variables for validation.

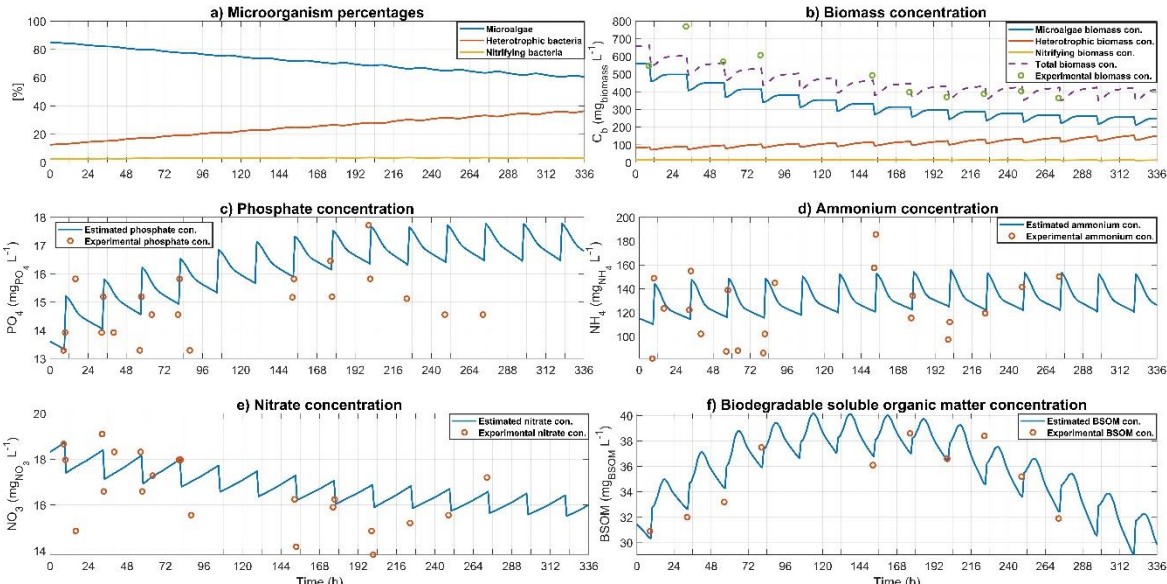

**Figure 7.** Validation results for the biomass production model with wastewater medium. (**a**) Microalgae and bacteria percentages inside the reactor; (**b**) Microalgae and bacteria biomass concentration; (**c**) Phosphate concentration inside the reactor; (**d**) Ammonium concentration inside the reactor; (**e**) Nitrate concentration inside the reactor; (**f**) Biodegradable soluble organic matter concentration inside the reactor.

Regarding errors, for the total biomass concentration, an error of 0.077 was obtained, a value almost identical to the result obtained in calibration. The error obtained for phosphate was 1.33, higher than the result obtained in the calibration. The error obtained for ammonia was 25.89, also higher than the result obtained in calibration. The error for nitrate was 1.7, slightly higher than the calibration result. Finally, the error obtained for the BSOM was 1.25, lower than the result obtained in calibration.

These results, at a preliminary level, show a good trend in the estimation of the elements of the model. Certain discrepancies in the results, as in the case of phosphate, may have been due to approximations and considerations in the input parameters of the model, such as the concentrations of the nutrients in the dilution medium, which change over time and have been considered constant.

More details about the parameters used in ABACO model can be found in Appendix A.

### 3.6. Respirometric Measurements

The respirometric measurements allowed us to determine the microalgal photosynthesis rate, the heterotrophic respiration rate, and the nitrifying respiration rate in the cultures. The microalgal photosynthesis rate was $15.8 \pm 2.3$ mgO$_2$ L$^{-1}$ h$^{-1}$, the heterotrophic respiration rate was $2.2 \pm 0.8$ mgO$_2$ L$^{-1}$ h$^{-1}$, and the nitrifying respiration rate was $0.27 \pm 0.1$ mgO$_2$ L$^{-1}$ h$^{-1}$. These values correspond to 86.7% microalgae, 11.8% heterotrophic bacteria, and 1.5 % nitrifying bacteria. These values closely approximate those determined by calibration (82.1% for microalgae, 13.2% for heterotrophic bacteria, and 4.7% for nitrifying bacteria) and validation (85% for microalgae, 12.6% for heterotrophic bacteria, and 2.4% for nitrifying bacteria) processes (Figure 8).

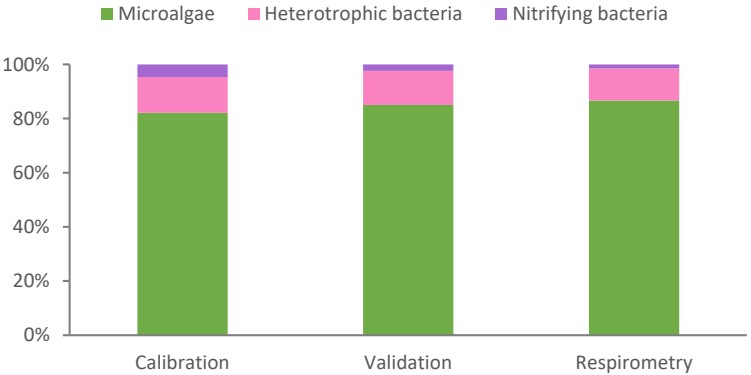

**Figure 8.** The microbial percentages obtained during the calibration process, the validation process, and the experimental respirometric measurements.

### 3.7. Discussion

The combination of microalgae biomass production processes and wastewater (diluted pig slurry) treatment is a cost-effective goal that poses several challenges. On the one hand, the already used wastewater contains high amounts of nutrients that allow microalgae and bacteria growth [27]. On the other hand, the lower energy demand for the microalgae–bacteria wastewater treatment, along with the ability of microalgae for CO$_2$ fixation, significantly increases the environmental sustainability of this eco-friendly technology [28]. Apart from multiple biological models proposed for microalgae–bacteria wastewater treatment using different types of effluents, scarce information is available about the use of animal manure as a nutrient source [29]. In this work, an integral microalgae and bacteria model named ABACO was developed, calibrated, and validated with experimental data from duplicate laboratory-scale photobioreactors using pig slurry as a nutrient source. The implementation of the model allowed us to simulate the dynamics of different components in the system and the relative proportions of microalgae and bacteria. The values of several model parameters were calibrated using genetic algorithms. Addi-

tionally, the percentage of each microbial population present in the microalgae–bacteria culture was estimated.

The results obtained from the comparison between the estimated values with respect to the measured experimental data have been satisfactory. The concentrations of the elements were adjusted within the range formed by the measurement points, despite being scattered data. The percentages of microalgae and bacteria within the reactor over time showed values close to those obtained in the literature [17,29]. Additionally, these percentages were estimated by a respirometric method, in which the microalgae, heterotrophic bacteria, and nitrifying bacteria showed estimated values that were close to those determined by calibration and validation processes. When analyzing the errors obtained for both cases, it becomes clear that faithfully estimating the evolution in the concentration of the different elements in the reactor is a complex process. The experimental values are very scattered, and that hinders their continuous evolution estimation. Even so, the results obtained are within the ranges of variation of the measurements taken.

The complexity in measuring individual concentrations of each species highlights the need for a reliable estimation method. Due to the calibration using genetic algorithms, it is possible to estimate the percentage of each microorganism in the reactor. From experimental data, the model allows one to determine the initial percentage of each element and estimate its evolution over time. In this way, the model can act as a simulator to predict the behavior of organisms based on the concentrations of nutrients present in the reactor medium. This model and the calibration parameters obtained will serve as the basis for the development of simulation models where the production of microalgae biomass is combined with wastewater treatment.

## 4. Conclusions

The microalgae–bacteria model proposed has demonstrated itself to be a useful tool for understanding the microalgal–bacterial interaction in wastewater treatment. The calibration carried out by means of genetic algorithms opens the door to a simple method of adjusting the various parameters that make up the model, so that it can be recalibrated from experimental measurements of different medium and culture scenarios, since the concentrations of nutrients vary from one type of medium to another. Therefore, the model could be applied to different strains, both microalgae and bacteria, by recalibrating the parameters based on a set of experimental data. The next step is focusing on the validation of the biological model in large-scale photobioreactors in order to find the optimal conditions for wastewater treatment, nutrient recovery, and biomass production, thereby enabling the sustainability of the process.

**Author Contributions:** Conceptualization, A.S.-Z. and E.R.-M.; methodology, A.S.-Z. and E.R.-M.; software, A.S.-Z. and E.R.-M.; validation, A.S.-Z. and E.R.-M.; formal analysis, A.S.-Z. and E.R.-M.; investigation, A.S.-Z.; resources, J.L.G. and F.G.A.-F.; data curation, A.S.-Z. and E.R.-M.; writing—original draft preparation, A.S.-Z. and E.R.-M.; writing—review and editing, J.L.G. and F.G.A.-F.; visualization, E.M.G. and J.M.F.-S.; supervision, E.M.G.; project administration, E.M and J.M.F.-S.; funding acquisition, J.L.G. and F.G.A.-F. All authors have read and agreed to the published version of the manuscript.

**Funding:** This work has been partially funded by the following projects: DPI2017 84259-C2- 1-R (financed by the Spanish Ministry of Science and Innovation and EU-ERDF funds), the European Union's Horizon 2020 Research and Innovation Program under grant agreement number 727874 SABANA, and the PURASOL project CTQ2017-84006-C3-3-R (financed by the Spanish Ministry of Economy and Competitiveness). It was also supported by the Spanish Ministry of Education through the National FPU Program (grant number FPU16/05996).

**Informed Consent Statement:** Not applicable.

**Data Availability Statement:** The data presented in this study are available on request from the corresponding author. The data are not publicly available due to privacy.

**Acknowledgments:** This work has been partially funded by the following projects: DPI2017 84259-C2- 1-R (financed by the Spanish Ministry of Science and Innovation and EU-ERDF funds), the European Union's Horizon 2020 Research and Innovation Program under Grant Agreement No. 727874 SABANA and the PURASOL project CTQ2017-84006-C3-3-R (financed by the Spanish Ministry of Economy and Competitiveness). As well as being supported by the Spanish Ministry of Education through the National FPU Program (grant number FPU16/05996).

**Conflicts of Interest:** There are no potential financial interests or others that could be perceived as influencing the outcome of the research. No conflicts, informed consent, or human or animal rights issues are applicable. All the authors confirmed authorship of the manuscript and agreed to submit it for peer review.

## Appendix A

**Table A1.** Variables for the proposed ABACO model.

| Variables of the Biologic Models | |
|---|---|
| **Heterotrophic Bacteria** | |
| $\mu_{het,max}$ | Heterotrophic bacteria maximum growth rate |
| $T_{min}$ | Minimal heterotrophic bacteria temperature |
| $T_{max}$ | Maximum heterotrophic bacteria temperature |
| $T_{opt}$ | Optimum heterotrophic bacteria temperature |
| $pH_{min}$ | Minimal heterotrophic bacteria pH |
| $pH_{max}$ | Maximum heterotrophic bacteria pH |
| $pH_{opt}$ | Optimum heterotrophic bacteria pH |
| $K_{S,DO_2,HET}$ | Heterotrophic bacteria half-saturation constant for dissolved oxygen |
| $K_{S,NH_4,HET}$ | Heterotrophic bacteria half-saturation constant for N-NH$_4$ |
| $K_{S,PO_4,HET}$ | Heterotrophic bacteria half-saturation constant for P-PO$_4$ |
| $K_{S,BSOM,HET}$ | Heterotrophic bacteria half-saturation constant for biodegradable soluble organic matter |
| $Y_{con}\left[\frac{NH_4}{het}\right]$ | Ammonium consumption rate from heterotrophic bacteria |
| $Y_{con}\left[\frac{PO_4}{het}\right]$ | Phosphate consumption rate from heterotrophic bacteria |
| $Y_{gen}\left[\frac{BSOM}{het}\right]$ | BSOM generation rate from heterotrophic bacteria |
| $Y_{con}\left[\frac{BSOM}{het}\right]$ | BSOM consumption rate from heterotrophic bacteria |
| **Nitrifiying Bacteria** | |
| $\mu_{nit,max}$ | Nitrifiying bacteria maximum growth rate |
| $T_{min}$ | Minimal nitrifiying bacteria temperature |
| $T_{max}$ | Maximum nitrifiying bacteria temperature |
| $T_{opt}$ | Optimum nitrifiying bacteria temperature |
| $pH_{min}$ | Minimal nitrifiying bacteria pH |
| $pH_{max}$ | Maximum nitrifiying bacteria pH |
| $pH_{opt}$ | Optimum nitrifiying bacteria pH |
| $K_{S,DO_2,NIT}$ | Nitrifiying bacteria half-saturation constant for dissolved oxygen |
| $K_{I,DO_2,NIT}$ | Nitrifiying bacteria inhibition constant for dissolved oxygen |
| $K_{S,C,NIT}$ | Nitrifiying saturation half-constant for CO$_2$ |
| $K_{S,NH_4,NIT}$ | Nitrifiying bacteria half-saturation constant for N-NH$_4$ |
| $K_{S,PO_4,NIT}$ | Nitrifiying bacteria half-saturation constant for P-PO$_4$ |
| $Y_{con}\left[\frac{NH_4}{nit}\right]$ | Ammonium consumption rate from nitrifying bacteria |
| $Y_{gen}\left[\frac{NO_3}{nit}\right]$ | Nitrate generation rate from nitrifying bacteria |
| $Y_{con}\left[\frac{PO_4}{nit}\right]$ | Phosphate consumption rate from nitrifying bacteria |
| $Y_{gen}\left[\frac{BSOM}{nit}\right]$ | BSOM generation rate from nitrifying bacteria |

**Table A2.** Values for the proposed ABACO model's characteristic parameters.

| Microalgae Net Photosynthesis Rate | | | |
|---|---|---|---|
| **Parameter** | **Value** | **Units** | **Source** |
| $\mu_{alg,max}$ | 1.591 | $day^{-1}$ | Calibrated |
| $I_k$ | 168 | $\mu E \cdot m^{-2} \cdot s^{-1}$ | Sánchez-Zurano et al., 2020 |
| $n$ | 1.700 | - | Sánchez-Zurano et al., 2020 |
| $T_{min}$ | 3.400 | °C | Sánchez-Zurano et al., 2020 |
| $T_{max}$ | 49 | °C | Sánchez-Zurano et al., 2020 |
| $T_{opt}$ | 30 | °C | Sánchez-Zurano et al., 2020 |
| $pH_{min}$ | 1.800 | - | Sánchez-Zurano et al., 2020 |
| $pH_{max}$ | 12.900 | - | Sánchez-Zurano et al., 2020 |
| $pH_{opt}$ | 8.500 | - | Sánchez-Zurano et al., 2020 |
| $DO_{2,max}$ | 32 | $mg_{O_2} \cdot L^{-1}$ | Sánchez-Zurano et al., 2020 |
| $m$ | 4.150 | - | Sánchez-Zurano et al., 2020 |
| $m_{max,alg}$ | 0.010 | $day^{-1}$ | Calibrated |
| $m_{min,alg}$ | 0.276 | $day^{-1}$ | Calibrated |
| $I_{k,resp}$ | 134 | $\mu E \cdot m^{-2} \cdot s^{-1}$ | Sánchez-Zurano et al., 2020 |
| $n_{resp}$ | 1.400 | - | Sánchez-Zurano et al., 2020 |
| $K_{S,C,ALG}$ | $4 \cdot 10^{-3}$ | $mg_C \cdot L^{-1}$ | BIO_ALGAE |
| $K_{I,C,ALG}$ | 120 | $mg_C \cdot L^{-1}$ | BIO_ALGAE |
| $K_{S,NH_4,ALG}$ | 1.540 | $mg_N \cdot L^{-1}$ | Sánchez-Zurano et al., 2020. Under rev. |
| $K_{I,NH_4,ALG}$ | 571 | $mg_N \cdot L^{-1}$ | Sánchez-Zurano et al., 2020. Under rev. |
| $K_{S,NO_3,ALG}$ | 2.770 | $mg_N \cdot L^{-1}$ | Sánchez-Zurano et al., 2020. Under rev. |
| $K_{I,NO_3,ALG}$ | 386.600 | $mg_N \cdot L^{-1}$ | Sánchez-Zurano et al., 2020. Under rev. |
| $K_{S,PO_4,ALG}$ | 0.430 | $mg_P \cdot L^{-1}$ | Sánchez-Zurano et al., 2020. Under rev. |
| $Y_{con}\left[\frac{NH_4}{alg}\right]$ | 0.369 | $g_{NH_4} \cdot g_{alg}^{-1}$ | Calibrated |
| $Y_{con}\left[\frac{NO_3}{alg}\right]$ | 0.214 | $g_{NO_3} \cdot g_{alg}^{-1}$ | Calibrated |
| $Y_{con}\left[\frac{PO_4}{alg}\right]$ | 0.008 | $g_{PO_4} \cdot g_{alg}^{-1}$ | Calibrated |
| $Y_{gen}\left[\frac{BSOM}{alg}\right]$ | 0.148 | $g_{BSOM} \cdot g_{alg}^{-1}$ | Calibrated |
| **Heterotrophic Respiration Rate** | | | |
| **Parameter** | **Value** | **Units** | **Source** |
| $\mu_{het,max}$ | 1.235 | $day^{-1}$ | Calibrated |
| $T_{min}$ | 9 | °C | Sánchez-Zurano et al., 2020 |
| $T_{max}$ | 47 | °C | Sánchez-Zurano et al., 2020 |
| $T_{opt}$ | 36 | °C | Sánchez-Zurano et al., 2020 |
| $pH_{min}$ | 6 | - | Sánchez-Zurano et al., 2020 |
| $pH_{max}$ | 12 | - | Sánchez-Zurano et al., 2020 |
| $pH_{opt}$ | 9 | - | Sánchez-Zurano et al., 2020 |
| $K_{S,DO_2,HET}$ | 1.980 | $mg_{O_2} \cdot L^{-1}$ | Sánchez-Zurano et al., 2020 |
| $K_{S,NH_4,HET}$ | 0.500 | $mg_N \cdot L^{-1}$ | ASM |
| $K_{S,PO_4,HET}$ | 0.010 | $mg_P \cdot L^{-1}$ | ASM |
| $K_{S,BSOM,HET}$ | 20 | $mg_{BSOM} \cdot L^{-1}$ | ASM |
| $Y_{con}\left[\frac{NH_4}{het}\right]$ | 0.299 | $g_{NH_4} \cdot g_{het}^{-1}$ | Calibrated |
| $Y_{con}\left[\frac{PO_4}{het}\right]$ | 0.017 | $g_{PO_4} \cdot g_{het}^{-1}$ | Calibrated |
| $Y_{gen}\left[\frac{BSOM}{het}\right]$ | 0.153 | $g_{BSOM} \cdot g_{het}^{-1}$ | Calibrated |
| $Y_{con}\left[\frac{BSOM}{het}\right]$ | 0.478 | $g_{BSOM} \cdot g_{het}^{-1}$ | Calibrated |

**Table A2.** *Cont.*

| Nitrifying Respiration Rate | | | |
|---|---|---|---|
| **Parameter** | **Value** | **Units** | **Source** |
| $\mu_{nit,max}$ | 0.730 | $day^{-1}$ | Calibrated |
| $T_{min}$ | 0 | °C | Sánchez-Zurano et al., 2020 |
| $T_{max}$ | 49 | °C | Sánchez-Zurano et al., 2020 |
| $T_{opt}$ | 33.600 | °C | Sánchez-Zurano et al., 2020 |
| $pH_{min}$ | 2 | - | Sánchez-Zurano et al., 2020 |
| $pH_{max}$ | 13.400 | - | Sánchez-Zurano et al., 2020 |
| $pH_{opt}$ | 9 | - | Sánchez-Zurano et al., 2020 |
| $K_{S,DO_2,NIT}$ | 1.080 | $mg_{O_2} \cdot L^{-1}$ | ASM |
| $K_{I,DO_2,NIT}$ | 104.900 | $mg_{O_2} \cdot L^{-1}$ | ASM |
| $K_{S,C,NIT}$ | 0.500 | $mg_C \cdot L^{-1}$ | ASM |
| $K_{S,NH_4,NIT}$ | 1 | $mg_N \cdot L^{-1}$ | ASM |
| $K_{S,PO_4,NIT}$ | 0.010 | $mg_P \cdot L^{-1}$ | ASM |
| $Y_{con}\left[\frac{NH_4}{nit}\right]$ | 3.224 | $g_{NH_4} \cdot g_{nit}^{-1}$ | Calibrated |
| $Y_{gen}\left[\frac{NO_3}{nit}\right]$ | 0.355 | $g_{NO_3} \cdot g_{nit}^{-1}$ | Calibrated |
| $Y_{con}\left[\frac{PO_4}{nit}\right]$ | 0.182 | $g_{PO_4} \cdot g_{nit}^{-1}$ | Calibrated |
| $Y_{gen}\left[\frac{BSOM}{nit}\right]$ | 0.149 | $g_{BSOM} \cdot g_{nit}^{-1}$ | Calibrated |

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
