# Peer review of "ABACO: A New Model of Microalgae-Bacteria Consortia for Biological Treatment of Wastewaters"

_applsci, doi:10.3390/app11030998_

Round 1

Reviewer 1 Report

The paper was clearly-written. In particular, there is ample description of the methodology. 

I would like to suggest that the authors include a quantitative measure of model accuracy (e.g. R2, mean absolute error, etc) aside from comparing the observed and estimated values. 

In the discussion, it was mentioned that the model accuracy was satisfactory. Perhaps this claim can be supported with quantitative measures, and with a comparison of the most similar work in literature. 

Author Response

Thank you very much for the comments and suggestions. We agree with you about a quantitative measure of the estimations. The cost function used in the calibration process has been the Root Mean Square Error (RMSE) between the estimated values ​​and those measured experimentally. The results of this error, both the total sum, and the individual errors of each variable (total biomass, phosphate, ammonium, nitrate and BSOM) have been included in the manuscript. Furthermore, this cost function has been used in the validation to quantitatively compare the results obtained with the calibration results data.

Reviewer 2 Report

   The manuscript provide a useful model to maximize biomass production , optimize the treatment capacity and provide more information of microalgae-bacteria wastewater treatment.   However, there are still some points needed to clarify  for a better comprehension. 

  First of all, the model was developed based on the species Scenedesmus almeriensis, if the model can applied with different microalgal species? Different microalgal species may have different requirement on the nutrient.For example, Dunaliella tertiolecta prefer nitrate than ammonium with the medium, Chlorella Vulgaris can also uptake organic nitrogen. On the other hand, if the model can applied with different types of wastewater such as food waste leachate which containing high organic nitrogen content?

  In the respiration rates determination, did it consider the respiration rate from other microorganisms such as zooplankton, fungi, or even the anaerobic microorganism exist in the dead zone of the bioreactor, etc?

 The experiments were conducted in continuous cultivation mode, if the dilution rate  affect the model development?

  In the validation of the model, it is found that most of the experimental results showed a great deviation withe the stimulated one. For example, the phosphate concentration in the later phase showed a descending trend while the stimulated one showed a stationary trend. The ammonium concentration in the middle phase increased to the highest concentration while the stimulated one showed a stationary trend. Do the model developed is still validated?

Lastly, if the biomass productivity can be stimulated by the model?

Author Response

Response to Reviewer #2

The manuscript provides a useful model to maximize biomass production, optimize the treatment capacity and provide more information of microalgae-bacteria wastewater treatment.   However, there are still some points needed to clarify for a better comprehension.

Thank you very much for the comments and suggestions.

 (1) First of all, the model was developed based on the species Scenedesmus almeriensis, if the model can applied with different microalgal species? Different microalgal species may have different requirement on the nutrient. For example, Dunaliella tertiolecta prefer nitrate than ammonium with the medium, Chlorella Vulgaris can also uptake organic nitrogen. On the other hand, if the model can applied with different types of wastewater such as food waste leachate which containing high organic nitrogen content?

Thank you for the interesting questions. As the reviewers pointed out, the model has been developed using Scenedemus almeriensis as the microalgae strain, but the model is adaptable to the use of other microalgae strains. The only requirement would be to use the characteristic parameters of each strain. In the case of Dunaliella tertiolecta, the main equation that considers the use of nitrogen in the form of nitrate should be used as the main equation. In the case of Chlorella vulgaris, the equation for the presence of organic nitrogen, its saturation constant and the measured experimental values ​​of organic nitrogen should be entered to calibrate and validate the model.

The model is also valid when other types of wastewater are used, in this case it would be necessary to check if the calibrated parameters allow a good fit of the model, if not, the model should be adapted. In the same way that has been said before, if it is necessary to consider organic nitrogen, it must be measured and introduced into the model. The most relevant fact is that the structure of the model is suitable to be used to whatever microalgae strain, only the specific values of the characteristics parameters of the model must be experimentally determined whatever the strain. Moreover, the already used methodology allows to determine this experimental values.

 (2) In the respiration rates determination, did it consider the respiration rate from other microorganisms such as zooplankton, fungi, or even the anaerobic microorganism exist in the dead zone of the bioreactor, etc?

We have considered in the model the presence of the three majority populations in the treatment of wastewater with microalgae. However, the model admits improvements and advances that we can incorporate step by step. For example, ASM biological models consider the presence of quite a few types of bacteria populations such as denitrifying bacteria or phosphorous accumulating bacteria (PAO). So, this work is an initial starting point, which will allow us to incorporate improvements or accessories to the model to make it as accurate as possible.

Regarding the presence of fungi and zooplakton, in most cases it could be united to heterotrophic bacteria and consider the "heterotrophic biomass" in general. However, in the cultures already evaluated the presence of fungi or zooplankton was really low, then it was not included into the model. Again, the model can be enlarged in the future to include different scenarios.

(3) The experiments were conducted in continuous cultivation mode, if the dilution rate affect the model development?

The dilution rate has been considered in the model. For example, this has been used to carry out the balances of nutrients and microorganisms, since it has a considerable influence. It is included in the model as Qd.

 (4) In the validation of the model, it is found that most of the experimental results showed a great deviation withe the stimulated one. For example, the phosphate concentration in the later phase showed a descending trend while the stimulated one showed a stationary trend. The ammonium concentration in the middle phase increased to the highest concentration while the stimulated one showed a stationary trend. Do the model developed is still validated?

The experimentally measured results are very scattered, but the estimations are within the dispersion range. As the revisor indicates, some variables fit the trend worse than others, such as ammonium. Even so, the variation, taking into account the scale, is small and it is a fairly satisfactory preliminary result comparing all the variables analyzed. These discrepancies may be due to approximations and considerations that have been taken in the input parameters in the model. For example, the concentration of nutrients in the dilution culture medium varies over time but has been taken as a constant value during the dilution of the reactor. This fact is something to consider in future improvements and implementations of the model.

(5) Lastly, if the biomass productivity can be stimulated by the model?

Yes, the model calculates the growth rate at each instant (minutes), so it is possible to calculate the biomass productivity as follows:

where  [min-1] is the specific growth rate and  [g m-3] is the microalgae biomass concentration.

This parameter has not been represented because the objective was to estimate the concentration of the measured variables, in this case, the nutrients and the biomass concentration.